# Impact of *dhps* mutations on sulfadoxine-pyrimethamine protective efficacy and implications for malaria chemoprevention

Andria Mousa [1,2] ✉, Gina Cuomo-Dannenburg [2], Hayley A. Thompson[3], David J. Bell[4], Umberto D'Alessandro[5], Roly Gosling[1,6], Alain Nahum[7,14], Karen I. Barnes [8], Jaishree Raman [9,10], Lesley Workmann[8], Yong See Foo[11], Jennifer A. Flegg[11], Emma Filtenborg Hocke[12,13], Helle Hansson[12,13], Ana Chopo-Pizarro [1], Khalid B. Beshir[1], Michael Alifrangis[12,13], R. Matthew Chico [1], Colin J. Sutherland [1], Lucy C. Okell [2,15] & Cally Roper [1,15]

Sulfadoxine-pyrimethamine (SP) is recommended for perennial malaria chemoprevention in young children in high burden areas across Africa. Mutations in the dihydropteroate synthase (*dhps*) gene (437 G̲/540E̲/581 G̲) associated with sulfadoxine resistance vary regionally, but their effect on SP protective efficacy is unclear. We retrospectively analyse time to microscopy and PCR-confirmed re-infection in seven efficacy trials including 1639 participants in 12 sites across Africa. We estimate the duration of SP protection against parasites with different genotypes using a Bayesian mathematical model that accounts for variation in transmission intensity and genotype frequencies. The longest duration of SP protection is >42 days against *dhps* sulfadoxine-susceptible parasites and 30.3 days (95%Credible Interval (CrI):17.1-45.1) against the West-African genotype *dhps* G̲KA (437G̲-K540-A581). A shorter duration of protection is estimated against parasites with additional mutations in the *dhps* gene, with 16.5 days (95%CrI:11.2-37.4) protection against parasites with the east-African genotype *dhps* G̲E̲A (437G̲-540E̲-A581) and 11.7 days (95%CrI:8.0-21.9) against highly resistant parasites carrying the *dhps* G̲E̲G̲ (437G̲-540E̲−581G̲) genotype. Using these estimates and modelled genotype frequencies we map SP protection across Africa. This approach and our estimated parameters can be directly applied to any setting using local genomic surveillance data to inform decision-making on where to scale-up SP-based chemoprevention or consider alternatives.

Infants and young children are at highest risk of severe malaria, with 76% of all malaria deaths occurring in children under five[1]. Sulfadoxine-pyrimethamine (SP) is recommended by the World Health Organisation (WHO) in perennial malaria chemoprevention (PMC), aiming to reduce malaria cases and deaths in young children[2]. PMC, which now includes the formerly known intermittent preventative treatment in infants (IPTi), involves administering a single SP dose to children without malaria symptoms at predefined intervals, in areas of moderate-to-high perennial transmission[2]. The latest WHO guidelines encourage tailored PMC implementation, allowing countries flexibility

to expand target age groups, adapt delivery platforms, drugs, and dosing schedules to site-specific resistance profiles, malaria endemicity, and seasonality[2]. A recent meta-analysis estimated a pooled efficacy of IPTi-SP across nine trials to be 22% against clinical malaria, 18% against anaemia and 15% against hospital admission[3]. However, only a few countries have adopted IPTi-SP, in part due to concerns about dosage and administration to infants, and perceived lack of protective efficacy.

An important determinant of antimalarial therapeutic and protective efficacy is the prevalence of mutations associated with drug resistance[4–7]. SP resistance is conferred by mutations in the dihydropteroate synthase (*Pfdhps*) and dihydrofolate reductase (*Pfdhfr*) genes and these are associated with clinical and parasitological drug failure[8]. The *dhfr* genotype IRN (51I-59R-108N), which is associated with partially reduced efficacy of pyrimethamine, spread quickly following the change of first-line treatment to SP and prevalence of these mutations remain high across sub-Saharan Africa[9,10]. In contrast, the prevalence of mutations in the *dhps* gene, which affect the sulfadoxine component of SP, varies by country and region[9]. In West and Central Africa, the *dhps* GKA (437G-K540-A581) in combination with *dhfr* IRN genotype confers partial resistance to SP[10]. In East Africa, the combination of *dhfr* IRN mutant with the *dhps* genotype GEA (437G-540E-A581), has been associated with SP treatment failure[8]. The further addition of *dhps* A581G generates the genotype GEG (437G-540E-581G), present in limited geographical foci in East and Southern Africa[11], and thought to confer even higher SP resistance[12,13].

Trials of IPTi and IPT in pregnant women (IPTp) suggest reduced effectiveness in areas with high prevalence of SP resistance markers, though data on resistance markers typically come from separate studies. In Korogwe, Tanzania (2004-2008), there was no significant protective efficacy conferred by IPTi-SP after 21 days[12,14,15]. Evidence from a trial done in the same area and period as the IPTi trial, showed that the GEG genotype was present in approximately half of the samples collected on the day of enrolment[13]. Similarly, IPTp-SP studies reported an association between the *dhps* GEG genotype and loss of protection from infection[16,17], and low birthweight[18]. An IPTi trial conducted in Uganda, reported an SP efficacy of just 7% against clinical malaria compared to the control arm. In the same district, a trial found almost all episodes of malaria on the day of recruitment were with parasites carrying the GEA genotype with no 581G[19]. However, IPTi-SP efficacy was sustained in a trial conducted in Maputo, Mozambique[20], where approximately half of the parasites harboured the *dhps* GEA genotype and half were *dhps* AKA (sulfadoxine-susceptible)[21]. In the absence of the 540E and 581G mutations, SP protective efficacy appears higher. In an area of low SP resistance in Navrongo, Ghana (2000–2002), a significant protective efficacy of IPTi with SP was estimated for the first 42 days following the last IPTi-SP dose[14,22,23]. The protective efficacy of SP is affected by the frequency of mutation-carrying parasites, but the exact impact of different *dhps* mutant combinations has never been systematically characterised.

Understanding the effect of different combinations of *dhps* mutations on the length of protection is important to inform chemoprevention strategies with SP, and SP-containing antimalarials, such as SP with amodiaquine (SPAQ)[2]. With renewed interest in PMC-SP adoption, national malaria programmes require evidence of protective efficacy in the presence of different resistance profiles to inform decisions on where PMC-SP can be implemented. Historical therapeutic efficacy studies (TES) of SP (or SP with artesunate, SPAS) with *dhps* and reinfection genotyping can be used to provide insights into the length of protection offered by SP against each genotype. Individually, these TES are not powered to assess protective efficacy and genotype effects. Here, we pool individual-level data from seven therapeutic efficacy studies in 1639 patients with a *Plasmodium falciparum* infection, collected from Malawi, Tanzania, Benin, Mozambique and South Africa, where new infections and detailed genotype data

were reported[13,24–30]. We quantify SP protective efficacy and mean duration of protection against new infections with each of the main *dhps* genotypes that are common in Africa: *dhps* AKA(*dhps* sulfadoxine-susceptible), GKA, GEA, and GEG, by fitting to trial data using an existing modelling framework that accounts for the underlying risk of infection and underlying genotype frequencies[31]. Further, we validate these findings using data from IPTi-SP studies.

## Results

### Study characteristics

We systematically searched the Worldwide Antimalarial Resistance Network (WWARN) Clinical Trials Publication Library to identify trials of SP or SPAS efficacy using PCR methods to distinguish new infections from recrudescent infections. A total of seven eligible studies were identified across 12 sites, in Malawi, Northern Tanzania, Benin, Mozambique and South Africa[13,24–28,30] between 2000 and 2006 (Table 1). The 1639 participants included in these studies were symptomatic malaria patients, consisting of children under five years for the studies conducted in Malawi, Northern Tanzania and Benin, or any ages >1 or >2 years for the studies in Mozambique and South Africa, respectively. Treatment failures were identified via positive blood smears at follow-up visits and classified as recrudescence or reinfection using PCR (Supplementary Table 1). Individual-participant data on time to new infection with each genotype since the time of drug administration were either obtained directly from the research groups (six studies) or extracted from publications where unavailable (one study)[25]. Data from a total of 21 trial arms were available, including SP, SPAS, SPAQ, chloroquine (CQ) and SPCQ. Across sites, all *dhps* parasite genotypes (AKA, GKA, GEA, GEG) were found (Fig. 1 and Table 1). Genotyping information on day 0 before SP treatment and on the day of failure was extracted where available. However, many samples contained missing information on the full *dhps* genotype (Supplementary Table 2), but these were still included in the analysis.

### Duration of protection against different genotypes

Data from the identified efficacy trials of SP and SPAS were used to estimate the protection against the sulfadoxine-susceptible *dhps* AKA or mutant genotypes (GKA, GEA, GEG). We use a deterministic multi-strain model[31] to model the probability of new infection with each genotype by fitting Weibull survival curves to the reinfection data using Hamiltonian Monte Carlo (HMC) methods (Supplementary Note 1). We fit the model to all data across 12 trial sites simultaneously accounting for the site-specific underlying malaria incidence and genotype frequency in the parasite population. The mean duration of protection against new infection was calculated using the estimated shape and scale parameters of the Weibull survival curve (Supplementary Note 1). In our modelling approach we model partial protection, rather than using a step function where someone is either protected or not on a particular day. Partial protection is expressed as a probability of protection over time since dose and this curve is constant across settings, irrespective of transmission.

Analysing all trial sites and drug arms together (Fig. 2), our model was able to fit the data well, with model-predicted values all within the 95% confidence intervals of the data with one exception of the SP arm in Malawi. Additionally, the model-predicted values for the genotype frequencies in each site closely matched the frequencies observed on day 0 (Supplementary Fig 1). Protection against sulfadoxine-susceptible parasites (*dhps* AKA) was significantly longer (55.7 days, 95% Credible Interval (CrI): 46.9–71.6) compared to most *dhps* mutant genotypes (Table 2, Supplementary Fig. 2 and Supplementary Table 3). The *dhps* GKA mutant reduced the duration of protection to 33.9 days (95% CrI: 16.8–56.8, $p = 0.143$), and the GEA mutant reduced protection further to 10.7 days (95% CrI: 8.9–21.9, $p < 0.001$). SP protection against the highly resistant *dhps* GEG genotype) was estimated to be similar to

**Table 1 | Summary of included studies**

| Publication | Site, Country, Year | Drug arms (N) | Follow-up (days) | Age of participants (mean) | *dhps* resistance profile* |
|---|---|---|---|---|---|
| Bell et al., 2008 | Blantyre, Malawi, 2003–2005 | SP (N = 114)<br>SPAS (N = 114)<br>SPAQ (N = 114)<br>SPCQ (N = 113) | 0, 1, 2, 3, 7, 14, 28 and 42 | 1–5 years (2.1 years) | 97.3% GEA<br>2.5% AKA<br>0.3% GKA |
| Gesase et al., 2009 | Tanga Region, Northern Tanzania, 2006 | SP (N = 87) | 0, 1, 2, 3, 7, 14, 21, and 28 | 6mo–5 years (2.2 years) | 50.0% GEA<br>45.7% GEG<br>4.3% AKA |
| Nahum et al., 2007; Nahum et al., 2009 | Cotonou, Benin, 2003–2005 | SP (N = 77)<br>SPAS (N = 81)<br>CQ (N = 79) | 0, 1, 2, 3, 7, 14, 21, 28, and those with ACPR/LPF were visited at home twice a week up to day 90. | 6mo-5 years (2.9 years) | ~85.0% GKA |
| Allen et al., 2009 | Magude, Mozambique, 2004–2005 | SP (N = 93)<br>SPAS (N = 86) | 0, 1, 2, 3, 7, 14, 21, 28, and 42 | all ages >1 year (15.2 years) | 88.5% AKA<br>11.5% GEA |
| | Boane, Mozambique, 2004–2005 | SP (N = 41)<br>SPAS (N = 63) | 0, 1, 2, 3, 7, 14, 21, 28, and 42 | all ages >1 year (20.5 years) | 82.6% AKA<br>17.4% GEA |
| | Namaacha, Mozambique, 2003 | SP (N = 40)<br>SPAS (N = 38) | 0, 1, 2, 3, 7, 14, 21, 28, and 42 | all ages >1 year (13.6 years) | 76.2% AKA<br>23.8% GEA |
| | Catuane, Mozambique, 2003 | SP (N = 24)<br>SPAS (N = 23) | 0, 1, 2, 3, 7, 14, 21, 28, and 42 | all ages >1 year (10.9 years) | 97.4% AKA<br>2.6% GEA |
| Barnes et al., 2006 | Namaacha, Mozambique, 2002 | SP (N = 97) | 0, 1, 2, 3, 7, 14, 21, 28, and 42 | all ages >1 year (14.9 years) | 90.0% AKA<br>5.7% GKA<br>4.3% GEA |
| | Bela Vista Mozambique, 2002 | SP (N = 49) | 0, 1, 2, 3, 7, 14, 21, 28, and 42 | all ages >1 year (11.2 years) | 70.5% AKA,<br>27.3% GEA,<br>2.3% GKA |
| | Bela Vista, Mozambique, 2003 | SP (N = 25) | 0, 1, 2, 3, 7, 14, 21, 28, and 42 | all ages >1 year (11.3 years) | 12.5% GEA<br>87.5% AKA |
| Barnes et al., 2008<br>Mabuza et al., 2005 | Mpumalanga, South Africa, 2002 | SP (N = 152) | 0, 1, 2, 3, 7, 14, 21, 28, and 42 | all ages >2 years (23.2 years) | 22.4% GEA<br>77.6% AKA |
| Bredenkamp et al., 2001 | Ndumu, KwaZulu-Natal, South Africa, 2000 | SP (N = 129) | 0, 1, 2, 3, 7, 14, 21, 28, and 42 | all ages >2 years (15.6 years) | ~90.0% GEA (based on data from Ndumu, 1999, published in Roper et al.[54].) |

*Frequencies estimated from unmixed day 0 infections in the data. If these were not available, statistics from the original publication are reported (unless indicating otherwise). The *dhps* AKA genotype indicates the sulfadoxine susceptible genotype with no *dhps* mutations. Gene names are shown in italics and mutations are underlined: *dhps* GKA (437G-K540-A581), *dhps* GEA (437G-540E-A581) and *dhps* GEG (437G-540E–581G).

the GEA mutant, at 11.7 days (95% CrI: 8.0–21.9, $p = 0.003$), but was based on only one study with shorter duration of follow-up.

SPAS provided a very similar duration of protection to SP against *dhps*- sulfadoxine-susceptible parasites (*dhps* AKA) and the *dhps* GKA mutant, as expected given the short elimination half-life of artesunate of <15 min[32]. However, SPAS provided significantly longer protection (16.5 days, 95% CrI: 11.2–37.4) against the *dhps* GEA genotype than did SP alone (10.7 days, 95% CrI: 8.9–21.9). The estimated 30-day protective efficacy of SPAS/SP against new infection in the 12 sites ranged between 15.4% to 98.1%, depending on the ratios of the genotypes present (Supplementary Table 4). The distribution of protective efficacy over time since treatment is shown in Fig. 3 for each genotype. Neither day 0 drug concentrations nor initial parasite density were associated with time to reinfection (Supplementary Note 2), though these were only available from a single trial[24,33].

We performed validation analysis against two IPTi trials conducted in Mozambique (2002–2004) and Tanzania (2004–2008). Using our model parameters estimated in the main analysis and the frequency of genotypes in the trial sites we predicted the mean duration of protection and the number of reinfections over time that would be expected in the IPTi trials following the first dose (Fig. 4). Accounting for weekly fluctuations in incidence without incorporating any genotype effects, the overall duration of protection offered by SP against clinical malaria was estimated to be 25.0 days (95% CrI: 12.0–41.5) and 10.9 days (95% CrI: 3.8–29.8) in the Mozambique and Tanzania IPTi trials, respectively. Similar estimates for protection were obtained when a time-constant force of infection was assumed across

follow-up (Supplementary Table 5). Withholding the control group from the analysis resulted in slightly shorter protection for the trial in Mozambique (20.7 days) and slightly longer protection for the trial in Tanzania (14.9 days). The expected duration of protection against any infection using the estimated genotype-specific parameters in the main analysis, allowing for the frequencies of genotypes in each site, was similar (22.1 days and 13.7 days for the Mozambican and Tanzanian IPTi trials, respectively). The model predictions closely follow the observed proportion of patients reinfected after the first SP dose in both trials (Fig. 4). In the IPTi trial in Mozambique, the model predicts that nearly all infections are *dhps* GEA over the first 30 days after SP. In the IPTi trial in Tanzania, a small number of new infections were observed between the first and second doses, with similar numbers in the SP and placebo arms, consistent with a short protection conferred by SP.

Trial data on other antimalarial therapies were included in the analysis (Table 2 and Supplementary Fig. 3). Despite SP showing a relatively short protection against *dhps* GEA, SPAQ showed a substantially longer protection of 42.5 days against this genotype, as expected given the long duration of action of amodiaquine's main active metabolite, desethylamodiaquine. This estimate is informed by trial data in Malawi, where the day 0 prevalence of *Pfcrt 76T*, *Pfmdr1 86Y* and *Pfmdr1 1246Y* mutations associated with amodiaquine resistance were low (0%, ~10% and 3%, respectively). In this study, there were no reinfections in the SPAQ arm by day 28 and only four reinfections were observed on day 42. In the same study, another SP-combination treatment, SPCQ, also showed a longer protection compared to SP (Table 2). Accounting for heterogeneity in the risk of

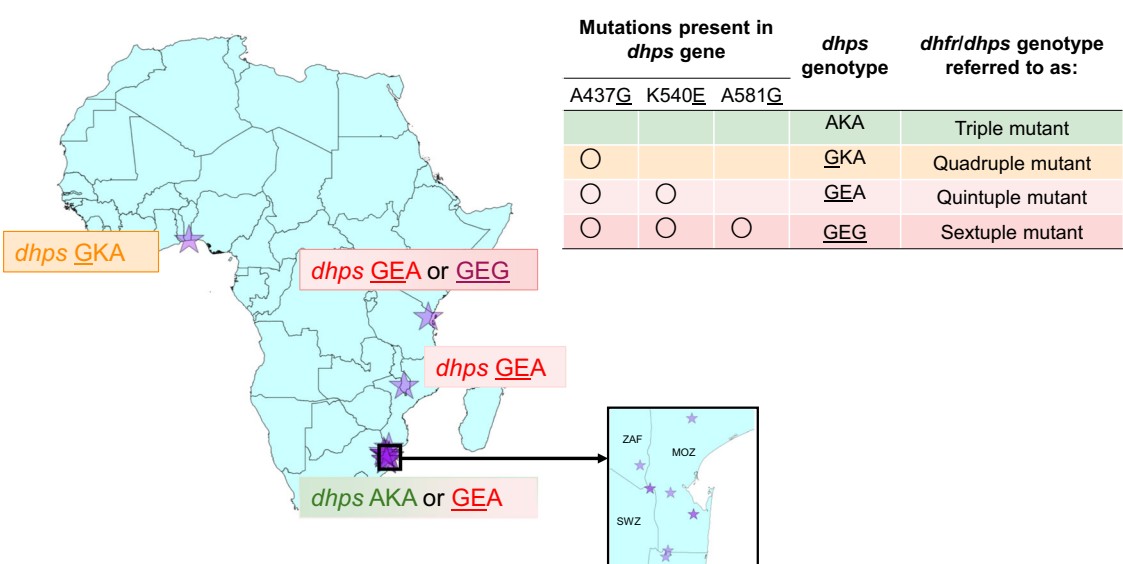

| Mutations present in *dhps* gene | | | *dhps* genotype | *dhfr/dhps* genotype referred to as: |
|---|---|---|---|---|
| A437G | K540E | A581G | | |
| | | | AKA | Triple mutant |
| ◯ | | | GKA | Quadruple mutant |
| ◯ | ◯ | | GEA | Quintuple mutant |
| ◯ | ◯ | ◯ | GEG | Sextuple mutant |

**Fig. 1 | Resistance genotype profile of the included studies.** Purple stars indicate the included sites. Annotations and the table denote the main *dihydropteroate synthase (dhps)* genotype profiles in those studies (with circles showing the presence of a mutation). The triple, quadruple, quintuple, and sextuple definitions indicated with the asterisk assume that the triple mutation in the *dihydrofolate reductase (dhfr)* gene (52I/ 59R/108N) is ubiquitous. The observed *dhps* GKA (437G-K540-A581) genotype may be considered as a proxy for the quadruple mutant, and the *dhps* GEA (437G-540E-A581) and GEG (437G-540E-581G) genotypes as proxies for the quintuple and sextuple mutants, respectively. ZAF=South Africa, MOZ= Mozambique, SWZ= Eswatini. Genotypes: *dhps* AKA (A437-K540-A581), *dhps* GKA (437G-K540-A581), *dhps* GEA (437G-540E-A581), *dhps* GEG (437G-540E-581G).

---

transmission resulted in similar estimates of mean duration of protection across all drugs (Supplementary Table 7).

### Applications in predicting impact of chemoprevention

Using our results on genotype-specific duration of SP protection, an SP protective efficacy prediction tool was developed which can be used to estimate PMC efficacy for any location where the frequencies of SP resistance markers are known: https://andriamousa.shinyapps.io/SP_PE_prediction_tool/ (Supplementary Note 3). In Fig. 5, applications of the tool are displayed for three example sites, each representing the main genotype profiles present in West Africa, East Africa and high-resistance pockets in East Africa.

Using our estimated protective efficacy profiles against each genotype, we used previously published estimates of *dhps* genotype frequencies[34] to predict the SP 30-day protective efficacy against any new infection and the median duration of protection (Fig. 6 and Supplementary Fig. 4). Across sub-Saharan Africa, 30-day protective efficacy varied from 59.3% to 91.5%, and the median duration of protection ranged from 17.2 to 37.2 days. We estimated that this protective efficacy corresponded to a mean of 4.7 clinical cases averted per 100 children aged 0 to 2 following a single-dose in areas of moderate-to-high transmission ($PRPf_{2-10} \geq 10\%$) (Fig. 7, Supplementary Fig 5). Similar reductions were estimated in the 0 to 5 year age group (Supplementary Fig 5).The highest mean number of clinical cases averted per 100 per dose was estimated for Liberia (8.9), followed by Benin (8.5), Sierra Leone (7.3) and Democratic Republic of the Congo (6.9).

### Discussion

Understanding the length of protection conferred by SP against new infections is paramount in informing and shaping effective chemoprevention policies. Our method allows estimation of the length of protection against different parasite genotypes, providing a valuable

tool for tailoring preventive strategies in diverse settings. Using reinfection data from therapeutic efficacy studies of SP or SPAS, we estimated a significantly shorter duration of SP protection against parasites carrying more mutations. Protective efficacy was maintained against sulfadoxine-susceptible parasites and those with only the *dhps* A437G mutation. However, parasites with the *dhps* GEA or GEG genotypes were associated with shorter durations of protection.

We observed a long duration of protection (56 days) against sulfadoxine-susceptible parasites, consistent with findings from a chemosensitivity study indicating a duration of in vivo inhibitory concentration of >52 days[35]. Evidence from an IPTi-SPAS trial conducted in a Senegalese setting where parasite genotypes were either *dhps* AKA (sulfadoxine-susceptible)(67%) or *dhps* GKA (29%) further supports the findings on the long duration of protection against sulfadoxine-susceptible strains[36]. In this trial, new infections in the month following the first SPAS dose occurred in 22% of the control cohort and in <2% of the intervention arm.

In the validation analysis, using IPTi trial data from two studies, one in a setting of *dhps* AKA (sulfadoxine-susceptible) genotype and the other of *dhps* GEA genotypes, we were able to replicate the trial results using genotype-specific parameters derived from our primary analysis. This underscores the reliability and generalizability of our findings, providing valuable validation for the application of estimated protection parameters to broader epidemiological contexts. However, the outcome in the TES analysis was patent infection, whereas the IPTi trial outcome was clinical infection. Any small differences between time to parasitaemia and time to clinical symptoms may not be captured between weekly follow-up visits. Furthermore, evidence from a systematic review suggest that efficacy against parasitaemia is not significantly different to efficacy against clinical infection[3]. However, more evidence is needed to assess whether protective efficacy against any infection is proportional to efficacy against clinical infection,

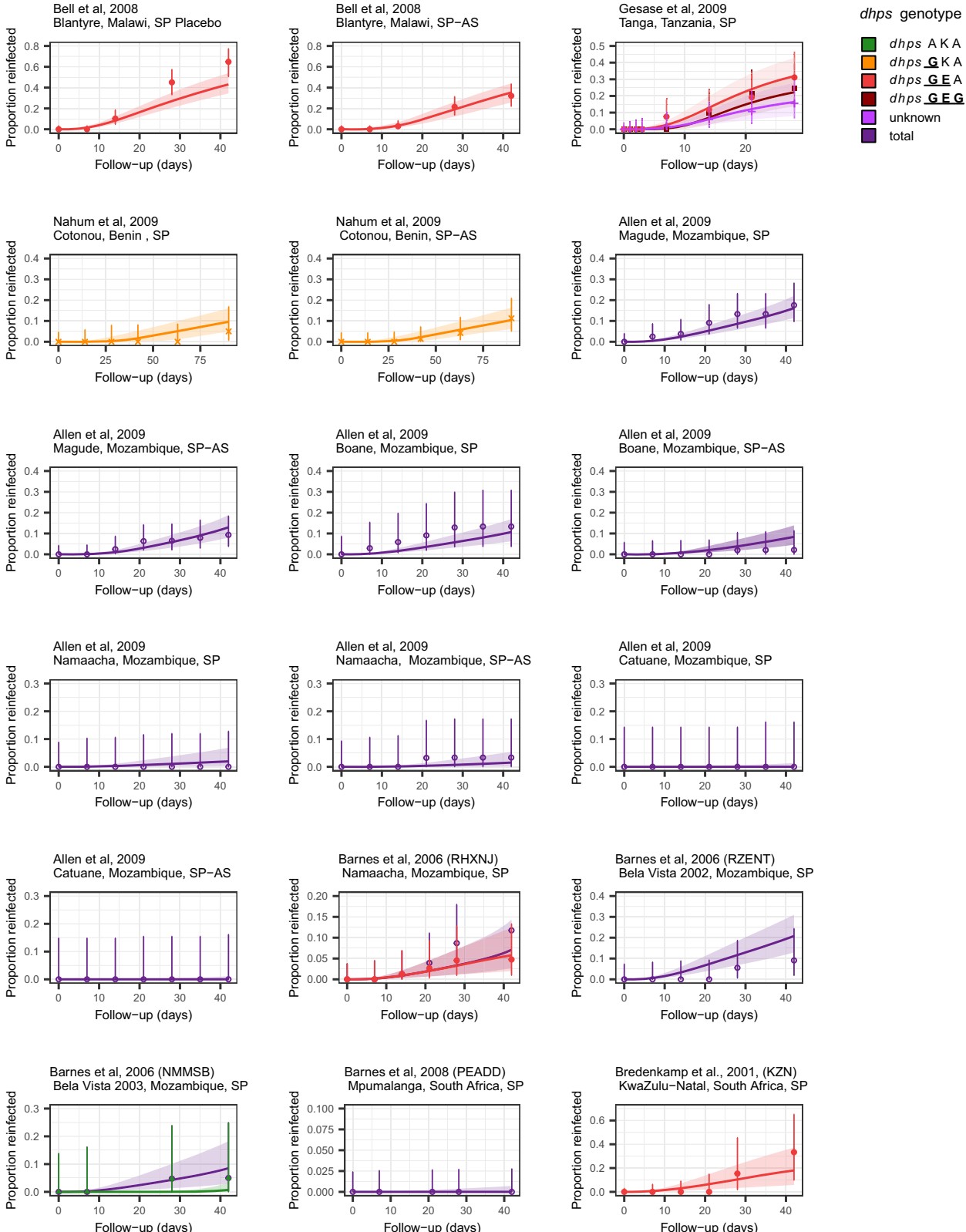

**Fig. 2 | Proportion of patients reinfected over time in each site and trial arm.** Markers denote the observed proportion infected across trial follow-up among those at risk of new infection, along with error bars representing 95% binomial confidence intervals (Clopper-Pearson method) around the observed proportion (*N* = 1639 across all trial arms and sites). The denominators for each trial site are shown in Table 1. Colours denote different genotypes. Model predictions from the combined fit are shown by the lines (posterior median) and shaded areas (95% Credible interval), *dhps* AKA (A437-K540-A581), *dhps* **G**KA (437**G**-K540-A581), *dhps* **GE**A (437**G**-540**E**-A581), *dhps* **GEG** (437**G**-540**E**-581**G**); *dhps*: dihydropteroate synthase.

**Table 2 | Model-estimated duration of protection by each drug against each *dhps* genotype**

| Drug group | *dhps* genotype (A437G/ K540E/A581G) | Mean duration of protection in days (Median and 95% Credible Interval) | *p*-value |
|---|---|---|---|
| Sulfadoxine-Pyrimethamine (SP) | AKA (sulfadoxine-susceptible) | 55.7 (46.9–71.6) | ref |
| | GKA | 33.9 (16.8–56.8) | 0.143 |
| | GEA | 10.7 (8.9–21.9) | <0.001 |
| | GEG | 11.7 (8.0–21.9) | 0.003 |
| Sulfadoxine-Pyrimethamine (SP) + Artesunate (SPAS) | AKA (sulfadoxine-susceptible) | 56.0 (46.8–72.1) | ref |
| | GKA | 30.3 (17.1–45.1) | 0.019 |
| | GEA | 16.5 (11.2–37.4) | 0.014 |
| Sulfadoxine-Pyrimethamine (SP) + Amodiaquine (SPAQ) † | GEA | 42.5 (36.7–52.4) | |
| Sulfadoxine-Pyrimethamine + Chloroquine (SPCQ) † | GEA | 23.8 (18.8–31.4) | |
| Chloroquine (CQ) | GKA | 27.1 (14.8–41.9) | |

*SP* sulfadoxine pyrimethamine, *SPAS* SP + artesunate, *SPAQ* SP + amodiaquine, *SPCQ* SP + chloroquine, *CQ* chloroquine †day 0 prevalence of *Pfcrt 76 T*, *Pfmdr1 86Y*, and *Pfmdr1 1246Y* mutations were low (0%, ~10%, and 3%, respectively) in the study conducted in Malawi.
Estimates are based on individual participant data with a total sample size of 1639 across 12 trial sites. *p*-values are calculated as the proportion of the posterior sample differences that are less than or greater than 0 (two-tailed test).The *dhps* AKA genotype indicates the sulfadoxine susceptible genotype with no *dhps* mutations. Gene names are shown in italics and mutations are underlined: *dhps* GKA (437G-K540-A581), dhps GEA (437G-540E-A581) and GEG (437G-540E-581G).

particularly in high-transmission areas with high rates of acquired immunity and highly prevalent asymptomatic parasitaemia.

In malaria-endemic regions characterized by high prevalence of SP-resistance-associated mutations, our study highlights the need for the strategic use of alternate regimens, such as SP plus Amodiaquine (SPAQ). SPAQ exhibits significantly higher chemoprevention efficacy compared to SP alone in areas where *dhps* GEA and GEG parasites are predominant. The long duration of protection of SPAQ is higher than SP or AQ alone, suggesting a potential boosting effect when co-administered[37]. To date, there is only one chemoprevention study using SPAQ in an area of saturated prevalence of *dhps* GEA[38]. However, in the study by Nuwa *et al*. there was a near-zero prevalence of mutations in the *chloroquine resistance transporter* (*crt*) and *multiple drug resistance 1* (*mdr 1*) genes that are associated with reduced efficacy of amodiaquine, similar to the TES in Malawi included here[24]. To allow extrapolation to other settings, analysis of SPAQ trial data from settings with a higher prevalence of these mutations is essential for understanding their impact on protective efficacy.

The need to use data from previous TES trials, not originally designed to estimate chemoprevention, adds to the uncertainty around the predicted duration of protection. Where AS was added to SP, we expect no added protection against parasites acquired after the drug dose compared to SP alone, as AS has a short half-life of <1 h[32,37]. However, based on published TES studies, the model predicted a shorter duration of protection for SP alone compared to SPAS. This is most likely due to misclassification errors whereby recrudescent episodes were scored as new infections following the drug dose in the TES data[39]. For instance, a parasite present on day 0 may be missed by microscopy, but then be detected during follow-up. In areas of high SP-resistance, this misclassification is more likely to occur in the SP group where more recrudescent infections are expected compared to the SPAS group[24,30], in which most day 0 infections will be cleared. Therefore, the duration of protection for SP alone may be under-estimated, and trials of SPAS may provide a more reliable estimate of protective efficacy. In future chemoprevention trials, analysing the risk of new infections in recipients who are parasite-negative at day 0 would solve this issue by removing cryptic recrudescent parasites as a source of misclassification.

One of the limitations of this study is the lack of control groups in the trial settings. This means the true underlying transmission rates are not directly measured and assumed to be constant. In the absence of a control group, a long follow-up is needed to isolate drug effects, and is particularly challenging in settings with fluctuating or seasonal transmission which can result in inaccurate estimates of protection[31]. Estimates for protection against the sulfadoxine-susceptible *dhps* genotypes were longer than the duration of follow-up in the studies where this genotype was present (42 days) so studies with a longer follow-up would be needed to confirm that. Nevertheless, the duration of SP protection is likely longer than 42 days, as supported by other studies[35]. Another limitation is that genotype data on the day of reinfection were limited for some of the studies and differences in the specific laboratory methods used by each study may also influence the findings. More data are needed to inform our parameter estimates, particularly on the more resistant *dhps* GEG genotype. This genotype was only covered by one of the included studies, which had a shorter follow-up of 28 days[13]. Other possible confounders such as the effect of the effect of age and immunity were not explored. Underdosing in young children is associated with treatment failure[40], and was not explored in this analysis. We expect that this effect is the same across studies, except in those using different age groups, where the genotype effects on protection may be harder to distinguish from age effects. However, our accurate prediction of the IPTi trial results despite differences in age groups, suggests that this confounding effect is minimal. Lastly, the analysis of historical studies can give no insight into protection against novel genotypes, such as *dhps*-431V which has emerged in West and Central Africa since SP was withdrawn as first-line treatment[41]. Furthermore, there was no individual-level data on the *dhfr*-164L mutation, though data by WWARN and a published literature review suggest that it is either absent or extremely rare in the countries of the included studies[9,42]. Quantifying the impact of the *dhfr*-164L mutation on protective efficacy would improve predictions when extrapolating to areas where these are present.

Quantifying the effects of genotype-specific protection is essential for modelling the suitability of SP in chemoprevention. This approach establishes a valuable methodology which can be applied across all epidemiological settings where resistance profiles have been characterised, and can be applied to other treatment regimens including SPAQ and DP, by quantifying the effects of relevant markers such as *mdr 1* and *crt*. This study highlights the need for molecular surveillance data to guide drug selection and roll out of PMC and other chemoprevention strategies. Integrating estimates of genotype-specific protective efficacy with molecular surveillance provides a robust foundation for evidence-based stratification of malaria chemoprevention across a range of transmission settings.

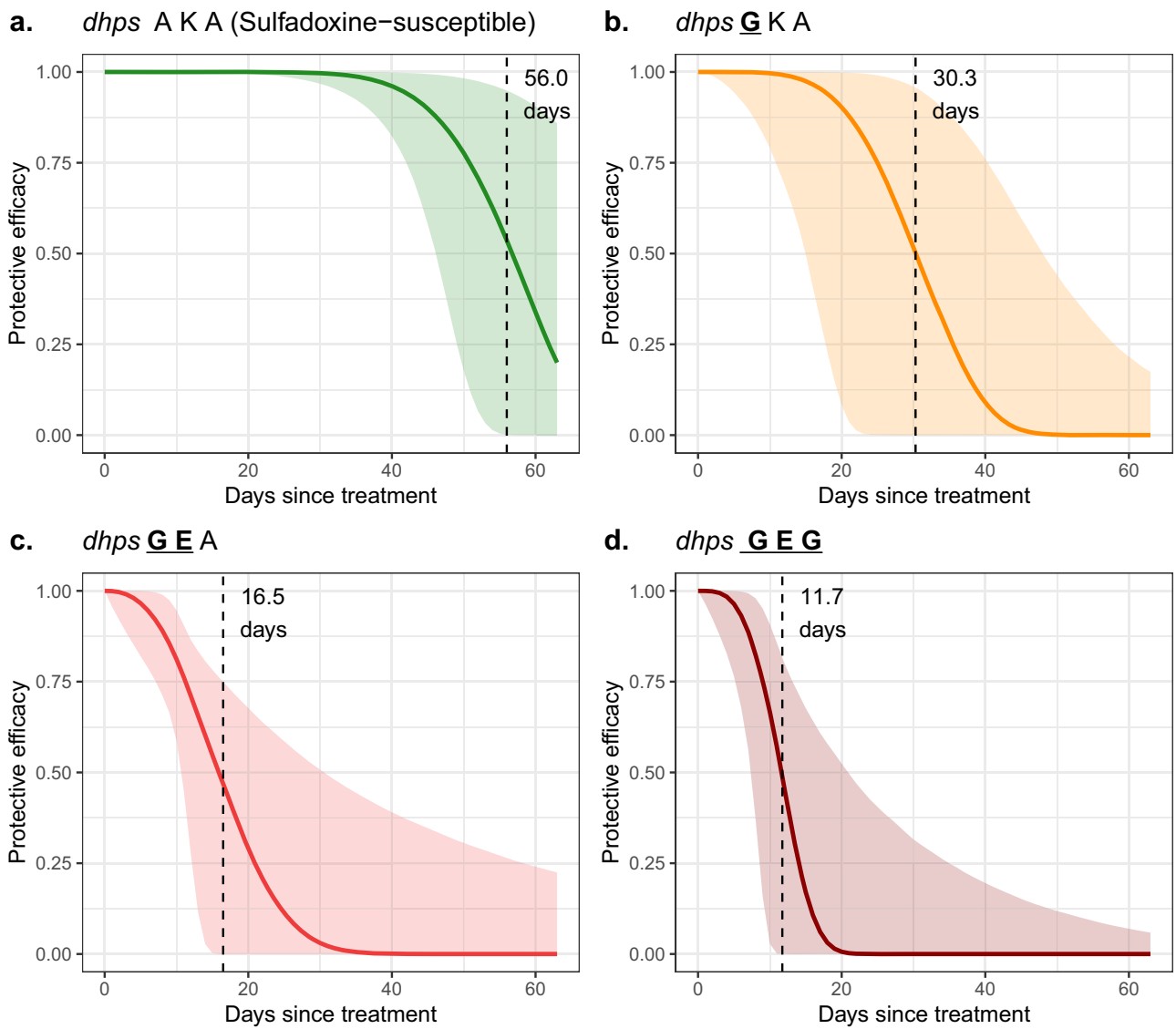

**Fig. 3 | Probability of protection by the drug (protective efficacy) since drug dose shown for each *dhps* genotype.** For each set of model parameters across 10,000 iterations, the probability of protection was calculated as a function of time since dose, and the Weibull shape and scale parameters. The solid lines denote the posterior median probability of protection and the shaded areas the 95% credible intervals. Mutations are underlined and shown in bold. Estimated parameters for

SPAS were used to obtain these curves, except for the *dhps* <u>G</u>E<u>G</u> genotype which uses SP-related parameters. The vertical line denotes the estimated mean duration of protection provided by the drug against each genotype. *dhps* AKA (A437-K540-A581) in panel (**a**), *dhps* <u>G</u>KA (437<u>G</u>-K540-A581) in panel (**b**), *dhps* <u>G</u>E<u>A</u> (437<u>G</u>-540<u>E</u>-A581) in panel (**c**), *dhps* <u>G</u>E<u>G</u> (437<u>G</u>-540<u>E</u>-581<u>G</u>) in panel (**d**); *dhps: dihydropteroate synthase*.

## Methods

### Data sources

We systematically screened SP and SPAS efficacy trials included in the WWARN Clinical Trials Publication Library[43]. Eligible studies (1) were conducted in Africa, (2) included SP or SPAS treatment groups, (3) applied polymerase chain reaction (PCR) methods on day 0 and day of failure samples to distinguish reinfection from recrudescence, (4) had a minimum of 42 days follow-up, and (5) collected genotype data on *dhps* mutations. SPAS trials were included because artesunate (AS) is a short-acting drug that provides no post-treatment prophylaxis[32,37]. No time of publication or age limits were applied, though studies of pregnant women were excluded due to confounding effects of immunity and multigravidity. Results from a previous analysis[31] indicated that a 28-day follow-up in single-arm trials may be insufficient to estimate the interval of protection. A follow-up of ≥42 days allows for more accurate disaggregation of drug protection and underlying transmission effects because the incidence is observed towards the end of follow-up when drug concentration levels are low. However, we

did include one study with 28 days follow-up as it was the only one where the highly resistant *dhps* <u>G</u>E<u>G</u> (437<u>G</u>-540<u>E</u>-581<u>G</u>) genotype was present[13].

We requested individual-participant data directly from research groups of identified studies. Data on time to infection with each genotype since the time of drug administration were either obtained from individual-participant data or, where unavailable[25], extracted from publications. Individuals were followed up after drug administration and monitored for treatment failure. All studies used PCR genotyping (*msp-2*, *msp-1* or *glurp* genes) to distinguish new infections from infections present on the day of drug administration (day 0) which had failed to clear or recrudesced (Supplementary Table 1). Recrudescent cases were censored on the day of failure when they received rescue treatment, except when new parasite variants were detected on the same day as the recrudescence. The presence of multiple parasite strains in a single sample in high transmission areas, may result in the majority sensitive genotype being detected but minority resistant infections being missed on day 0. For each

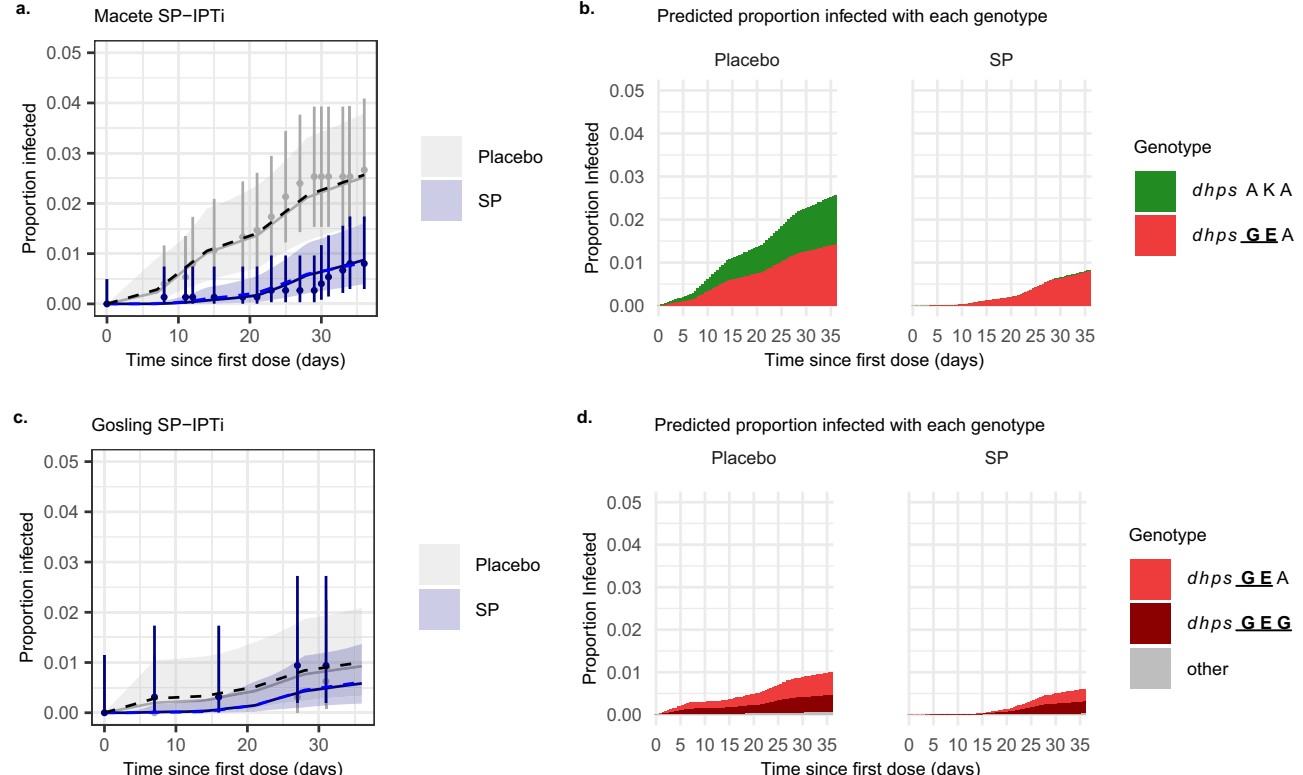

**Fig. 4 | Validation analysis using IPTi trial data.** All panels show the predicted proportion infected over time since the SP dose among those at risk of new infection for the Macete et al. trial[20] (**a**, **b**) and the Gosling et al. trial[12] (**c**, **d**). In panels **a** and **c**, dots represent the observed proportion of infections, with error bars denoting 95% binomial confidence intervals (Clopper-Pearson method) around the observed proportion. The sample size for plots **a** and **c** was 1497 for Mozambique (SP = 747 and placebo = 750) and 639 for Tanzania (SP = 319 and placebo = 320). Solid lines and shaded areas denote the model fit to the data (posterior median) and associated 95% Credible Intervals. Dashed lines show the predicted proportion infected given the estimated protection parameters from the main analysis and estimated frequency of each genotype (56% *dhps* GEA and 44% *dhps* AKA in the Mozambique trial[21], and 53.6% *dhps* GEA and 41.1% *dhps* GEG in Northern Tanzania[13]). Panels **b** and **d** show the predicted proportions of infection with each genotype for each treatment. **c** *dhps: dihydropteroate synthase*.

individual, the *dhps* genotypes present in the samples collected on day 0 were compared with those present on the day of failure. Where participants had a mixed infection on both the day of failure and the day of new infection, and where the *dhps* genotype of the new infection could not be distinguished from the original infection, the genotype of the new infection was considered *undetermined*, but was still analysed.

Drug concentration data on sulfadoxine and pyrimethamine on day 7 following drug administration were only partially available for one study[24,33]. For this study, we summarise the mean and median post-treatment drug concentrations on day 0 and day 7, along with initial parasite density by treatment outcome (new infection vs. no new infection during follow-up). In the Supplementary Note 2, we report results from a Cox-regression model on time to new infection accounting for day 0 drug concentrations and parasite density to explore the possibility of confounding effects.

**Analysis methods**

Trial data were used to estimate the protection against the four genotypes: *dhps* AKA (sulfadoxine-susceptible), GKA, GEA, GEG. We recently developed a deterministic multi-strain model describing new infection after treatment[31] which we used here to quantify SP protective efficacy, building on previous modelling approaches[44,45] (Supplementary Note 1). In brief, the probability of being protected by the drug was quantified at each time-step following treatment, by fitting Weibull survival curves to the reinfection data using Hamiltonian Monte Carlo (HMC) methods in RStan[46]. We fit the model to all data across 12 trial sites simultaneously estimating the site-specific underlying incidence of infection (assumed constant over study follow-up),

the site-specific frequency of each genotype in the parasite population, and independent protection curves against each genotypic strain. For studies with more than one treatment group, we fit different Weibull protection curves for each drug. Incidence of infection was assumed to be constant over the 42 days follow-up, due to the challenge of identifying fluctuating transmission effects in the absence of a control group.

30-day protective efficacy against first infection was estimated as the percentage of new infections with each strain prevented by the drug compared to a theoretical control group of no chemoprevention over 30 days. A single-strain model was used in places with limited genotype data and where drug resistance was high (prevalence of resistance genotype on day 0 > 85%), assuming that new infections consisted of resistant parasites. A two-strain model, that incorporates the frequency of each genotype, was used for studies where more than one genotype was present. The time-step used in the model (dt) was 0.5 days. We used relatively uninformative priors for all parameters related to drug protection effects and frequency of genotypes (Supplementary Table 3).

In the absence of a control group, without a reasonably informative prior for malaria incidence, the risk of infection is difficult to distinguish from the protective effect of the drug. Hence, priors used for malaria incidence in each site were semi-informative and were based on predictions using the Imperial College model of malaria transmission[47,48] calibrated to reported prevalence of parasitaemia where available[26,30]. If unavailable, we calibrated the model to predicted prevalence from the Malaria Atlas Project[49,50] specific to the year and place of the survey[13,24-29] (Supplementary Table 3 and Supplementary Fig 6). We ran 10,000 model iterations and four chains

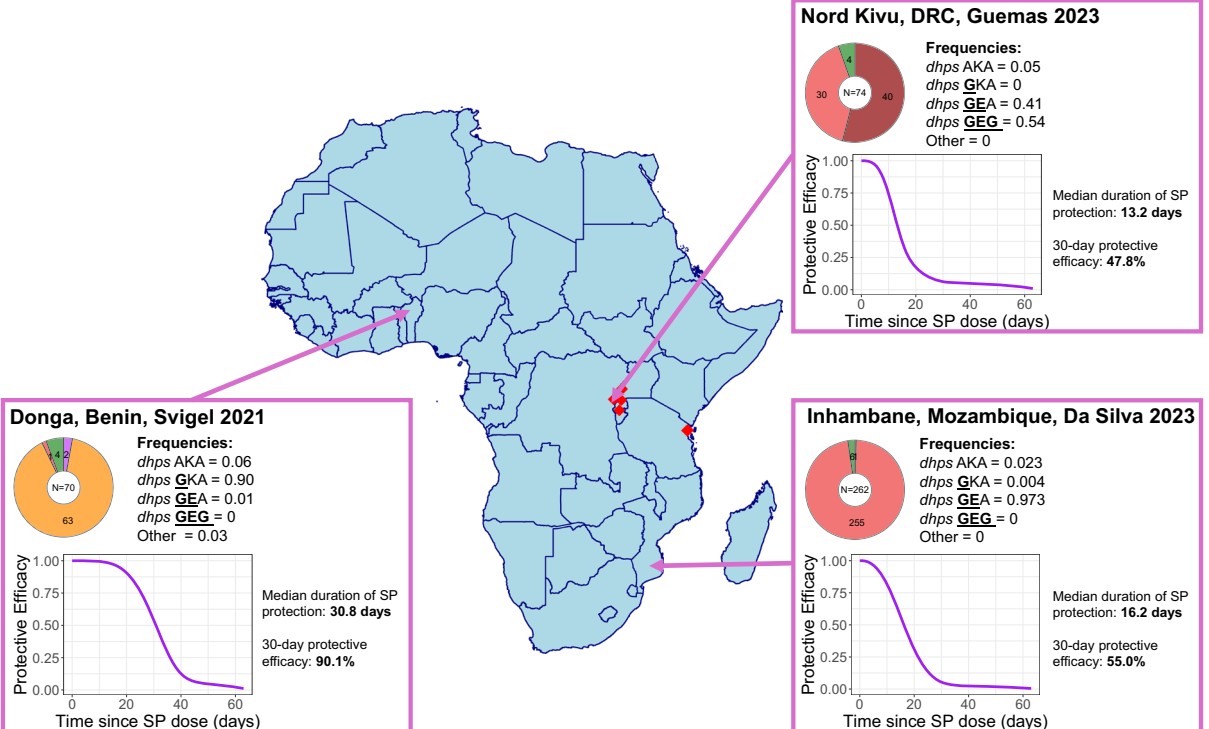

**Fig. 5 | Estimating protective efficacy and duration of protection for different sites based on genotype compositions.** Each insert uses published data on genotype prevalence for three sites in Donga, Benin[55], Nord Kivu, DRC[56], and Inhambane, Mozambique[57]. To estimate the 30-day protective efficacy, we used an incidence of malaria of 0.84, 0.15, and 0.29 infections per person per year for Donga, Nord-Kivu, and Inhambane, respectively. The *dhps* AKA is the sulfadoxine-susceptible genotype. Mutations are underlined and shown in bold. Red markers denote surveys conducted in Africa which reported a prevalence of >50% for *dhps* A581G and >90% for *dhps* K540E; *dhps* AKA (A437-K540-A581), *dhps* GKA (437G-K540-A581), *dhps* GEA (437G-540E-A581), *dhps* GEG (437G-540E-581G); *dhps*: dihydropteroate synthase, DRC: Democratic Republic of the Congo.

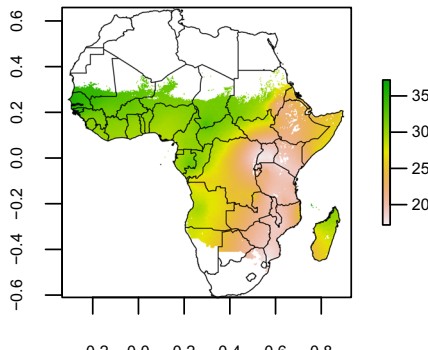

**Fig. 6 | Protection provided by sulfadoxine pyrimethamine across sub-Saharan Africa.** Estimated 30-day protective efficacy (panel a) and median duration of protection (panel b). These are based on the genotype-specific Weibull shape and scale parameters estimated in this analysis and the frequency of each genotype estimated in a previous study for 2020[34]. Protective efficacy in this figure incorporates reinfections (protective efficacy against any episode, rather than first episode). See Supplementary Note 3 for details on how these metrics were estimated. Coordinates are shown in radians.

(5000 burn-in iterations per chain). Convergence of all MCMC chains was assessed by visually assessing the posterior distributions and traceplots, and using a threshold of <1.05 for the Gelman-Rubin's convergence diagnostic ($\hat{R}$) and >1000 for the effective sample size (ESS) and effective tail distribution (Tail-ESS) per chain[51] (Supplementary Table 6). As a sensitivity analysis, we explored the effect of including heterogeneity in risk of transmission in the main analysis. To account for variation in risk between individuals, three risk groups were used, partitioned using Gaussian quadrature (Supplementary Table 7).

If the studies included additional drug arms other than SP/SPAS, these were also used in the model fitting to provide information on the underlying incidence of infection and genotype frequencies (Supplementary Fig. 3). Studies were analysed together, by fitting site-specific background malaria incidence and underlying genotype frequencies, and a pooled estimate of SP protection against each genotype (by drug arm). Using the parameter estimates derived from this analysis, we developed a simple web-based interactive tool to predict the chemoprevention efficacy of SP alone for areas of varying resistance and endemicity profiles (Supplementary Note 3).

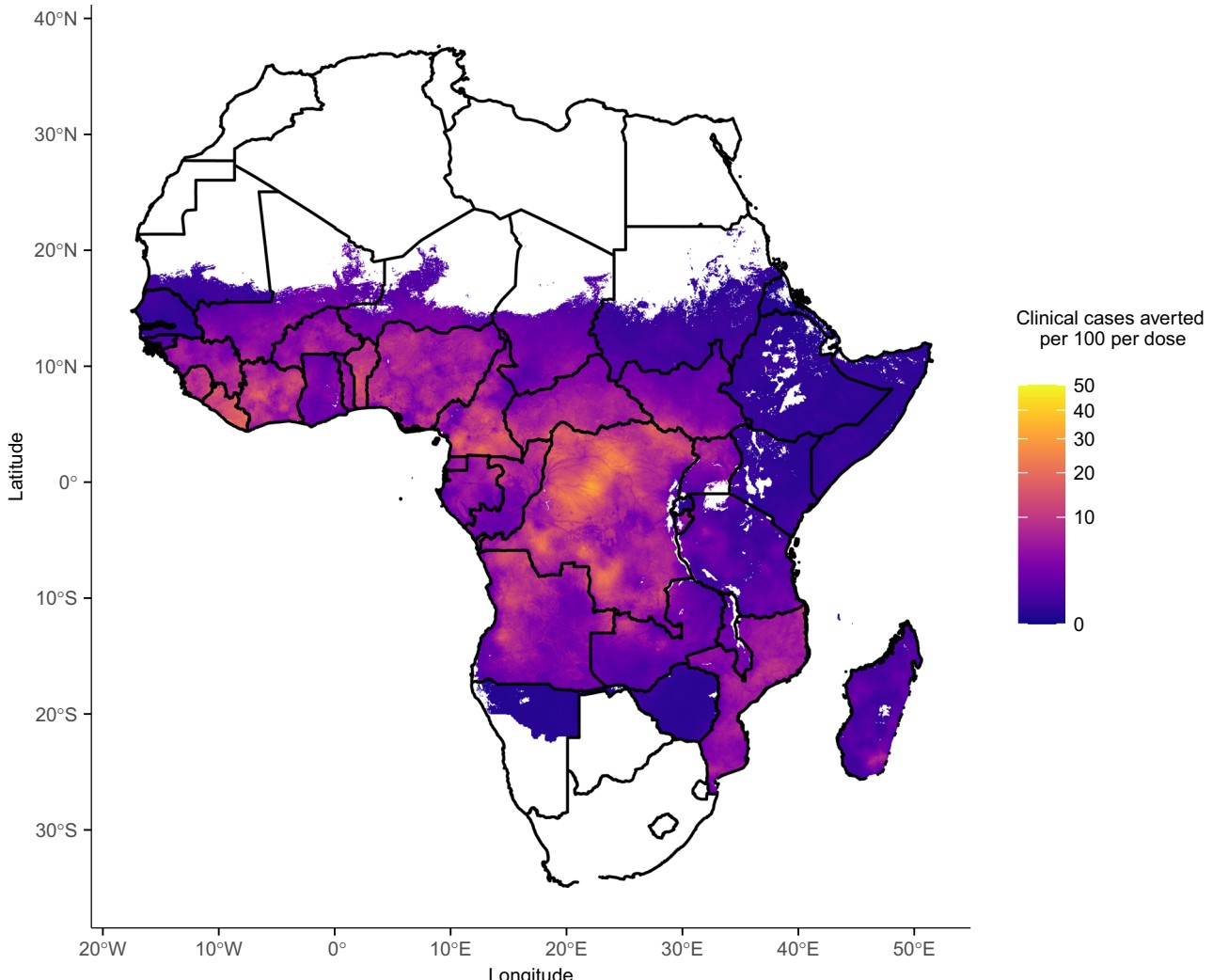

**Fig. 7 | Estimated clinical malaria cases per 100 averted following a single SP dose for ages 0 to 2 years.** These are based on prevalence rate estimates from the Malaria Atlas Project in 2–10 year olds ($Pf$PR$_{2\text{-}10}$) and the relationship between $Pf$PR$_{2\text{-}10}$ and incidence in children aged 0 to 2, obtained from the Imperial College malaria transmission model[47,48]. Incidence following a single SP dose was estimated using the 30-day protective efficacy shown in Fig. 6.

A recently published study estimated the frequency of *dhps* haplotypes across sub-Saharan Africa, using a Bayesian spatiotemporal model and molecular surveillance data from WWARN[34]. Using these estimates (Supplementary Fig. 4) we apply protective efficacy parameters obtained from the current analysis to predict SP chemoprevention impact across sub-Saharan Africa in 2020 (Fig. 6).

We used prevalence estimates in children aged 2 to 10 for the year 2020 obtained from the Malaria Atlas Project at 5 km × 5 km resolution[49]. The relationship between parasite prevalence in 2 to 10 year olds and incidence in 0 to 2 year olds, obtained from an agent-based malaria transmission model[47,48], was then used to infer incidence in each 5 km × 5 km pixel across Africa. The 30-day protective efficacy was then applied to these incidence metrics to estimate the number of clinical cases aged 0 to 2 averted per population after a single SP dose. The same process was repeated with incidence in 0 to 5 year olds to compare impact in these two age groups. To calculate mean clinical cases averted per dose for each country in areas of moderate to high transmission, we weighted the values by the population in each 5 km × 5 km, using 2020 WorldPop population data[52].

**Validation**
Placebo-controlled IPTi-SP trial data[12,20] were used for validation of our results, and were obtained from trial investigators or digitized from published Kaplan-Meier survival curves for both treatment and control arms after the first dose of SP in the trial and before any further doses. Where no *dhps* genotype data were available from the IPTi trial participants, we used data on the frequency of SP-resistance markers in that area and year from other sources. In the case of the IPTi trial in Mozambique[20], the genotype frequency information was obtained from samples collected from symptomatic malaria cases in the placebo group of the IPTi trial[21], representing population frequencies in the absence of drug selection. In the case of the IPTi trial in Tanzania, frequency estimates were obtained from samples collected in a health facility within 30 km of the IPTi implementing district, and during the same time as the IPTi trial (in 2006)[13]. Using both control and treatment arms, we estimated weekly clinical incidence and a Weibull survival curve for protective efficacy using RStan as above. We then predicted new infections given the estimated incidence and local frequencies of resistance genotypes[13,21] and compared the predicted duration of SP protection with that estimated in the observed IPTi data used for validation.

As an additional sensitivity analysis we repeated the validation analysis with a time-constant force of infection and compared the estimated incidence rate in the IPTi trial by including and excluding the placebo-controlled arm. We further compared the estimated duration of protection with that obtained from the analysis using time-varying

incidence, and that expected using the protection parameters from the main analysis of the TES trials.

All analyses and visualisations were performed in R version 4.0.3. All maps presented use national boundaries obtained from the global administrative areas database (GADM, version 4.1).

## Ethics statement

All studies included in this analysis have been published. Informed consent was obtained from participants or their parents or guardians. All individual studies were approved by both institutional ethics committees and local ethics review committees. This secondary analysis of trial data has been approved by the London School of Hygiene and Tropical Medicine ethics committee (reference: 29340).

## Reporting summary

Further information on research design is available in the Nature Portfolio Reporting Summary linked to this article.

## Data availability

The data that support the findings of this study are available with no restrictions in https://github.com/AndriaMousa/SP-resistance-protective-efficacy-code. The repository containing the data has been archived, with accession number: 10.5281/zenodo.14988819[53].

## Code availability

The analysis code that supports the findings of this study is available with no restrictions in https://github.com/AndriaMousa/SP-resistance-protective-efficacy-code. The repository containing the code has been archived, with accession number: 10.5281/zenodo.14988819[53].

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

## Acknowledgements

A.M., C.R., R.G., R.M.C., C.J.S., K.B.B., A.C.P., M.A., E.F.H. and H.H. were funded by Unitaid as part of the Plus Project (www.unitaid.org, grant number: 101150IC). G.C.D. and L.O. acknowledge funding from the UK Royal Society and from the MRC Centre for Global Infectious Disease Analysis (reference MR/R015600/1), jointly funded by the UK Medical Research Council (MRC) and the UK Foreign, Commonwealth & Development Office (FCDO), under the MRC/FCDO Concordat agreement and is also part of the EDCTP2 programme supported by the European Union. KIB acknowledges funding from the WHO TDR and The Global Fund to Fight AIDS, Tuberculosis and Malaria (GFATM). The funders had no role in the study design, data collection, analysis, interpretation, decision to publish, or preparation of the manuscript. We would like to acknowledge Ilona Carneiro for help in interpreting IPTi trial data in Tanzania. We would also like to acknowledge WWARN Oxford for the curation of data related to the therapeutic efficacy studies conducted in Mozambique and South Africa.

## Author contributions

A.M., C.R., L.C.O., C.J.S., R.M.C. and R.G. contributed to the conceptualisation of the study. A.M., C.R. and L.C.O. contributed to study design, and C.R. and L.C.O. supervised the work. A.M., C.R., L.C.O., G.C.D. and individual study data contributors had access to the data and have contributed to verifying underlying data and analyses. AM performed analyses reported in the manuscript, and GCD and HAT contributed to the code and methodology. D.J.B., U.D.A., R.G., A.N., K.I.B., J.R., L.W., J.A.F. and Y.S.F. contributed to data acquisition, data curation and cleaning and helped interpretation of the trial data used in this paper. A.M., G.C.D., H.A.T., D.J.B., U.D.A., R.G., A.N., K.I.B., J.R., L.W., Y.S.F., J.A.F., E.F.H., H.H., A.C.P., K.B.B., M.A., R.M.C., C.J.S., L.C.O., and C.R. assisted in the interpretation of data and results. AM wrote the first draft of the paper. A.M., G.C.D., H.A.T., D.J.B., U.D.A., R.G., K.I.B., J.R., L.W., Y.S.F., J.A.F., E.F.H., H.H., A.C.P., K.B.B., M.A., R.M.C., C.J.S., L.C.O., and C.R. reviewed and edited the paper for important intellectual content and approved the final paper for publication.

## Competing interests

All authors declare no competing interests as defined by Nature Portfolio, or other interests that might be perceived to influence the results and/or discussion reported in this paper.

## Additional information

[1]Faculty of Infectious and Tropical Diseases, London School of Hygiene and Tropical Medicine, WC1E 7HT London, UK. [2]MRC Centre for Global Infectious Disease Analysis, Department of Infectious Disease Epidemiology, Imperial College London, W12 0BZ London, UK. [3]Malaria and Neglected Tropical Diseases, PATH, WA 98121 Seattle, WA, US. [4]Department of Infectious Disease, NHS Greater Glasgow and Clyde, G51 4TF Glasgow, UK. [5]MRC Unit The Gambia at the London School of Hygiene and Tropical Medicine, PO Box 273 Serrekunda, Gambia. [6]Malaria Elimination Initiative, Institute of Global Health Sciences, University of California San Francisco, CA 94158 San Francisco, USA. [7]Centre de recherche entomologique de Cotonou, 2604 Cotonou, Benin. [8]Division of Clinical Pharmacology, Department of Medicine, University of Cape Town, 7925 Cape Town, South Africa. [9]South African National Institute for Communicable Diseases, 2192 Johannesburg, South Africa. [10]Wits Research Institute for Malaria, Faculty of Health Sciences, University of Witwatersrand, 2193 Johannesburg, South Africa. [11]School of Mathematics and Statistics, The University of Melbourne, 3052 Parkville, Australia. [12]Department of Immunology and Microbiology, Centre for translational Medicine and Parasitology, University of Copenhagen, 2200 Copenhagen, Denmark. [13]Department of Infectious Diseases, Copenhagen University Hospital, 2200 Copenhagen, Denmark. [14]Deceased: Alain Nahum. [15]These authors jointly supervised this work: Lucy C. Okell, Cally Roper. ✉e-mail: Andria.Mousa@lshtm.ac.uk

