## [Peer Review file · Nature Communications]

Impact of *dhps* mutations on sulfadoxine-pyrimethamine protective efficacy and implications for malaria chemoprevention

Corresponding Author: Dr Andria Mousa

Version 0:

Reviewer comments:

Reviewer #1

(Remarks to the Author)

In the manuscript entitled "Impact of *dhps* mutations on sulfadoxine-pyrimethamine protective efficacy and implications for malaria chemoprevention" the authors present the results of a statistical analysis to estimate the relationship between *dhps* mutation type and duration of protective efficacy against malaria re-infection. Overall the study is of high quality, by which I mean that the objective is valuable and due diligence and effort has been applied to the data preparation and model development; also, the statistical methods are consistent with best practice for hierarchical modelling in the contemporary literature. I have two comments/questions that I would urge the authors to address in their revisions. The first is critical to establishing the validity of the model-based approach, while the second is concerned with facilitating clear interpretation of results for policy makers.

The dataset used in this analysis come from a type of drug efficacy trial in which there is no untreated arm from which to estimate the background force of infection. For this reason the force of infection is treated as a latent model parameter that takes a constant value over the study duration at each site; this constant force of infection at each site is assigned a semi-informative prior in a Bayesian framework based on a mathematical model for the relationship between prevalence and the entomological inoculation rate. As acknowledged in the Discussion, the constant rate assumption is quite limiting, as any variations of transmission intensity during the study week will confound the shape estimates of the efficacy functions. I would guess that a further limitation is the assumption of no heterogeneity in the exposure risk of individuals; certainly the mathematic model used to estimate the priors for the force of infection at each site considers both heterogeneity between individuals and an age-dependence of the biting age. On the other hand, the comparison models fitted to the two validation datasets for which there are control arms are constructed with time-varying incidence rates, which makes the agreement between the output curves and the data difficult to assess. Further sensitivity analyses would therefore be appreciated. For instance, examining how the latent incidence rate estimates for the validation datasets compare with the observations when their control arms are withheld (under a time-constant force of infection model).

Regarding the presentation of final results, while the duration of protection metrics have a clear mathematical explanation I fear that they may not be meaningful to potential end-users in the policy sphere, since (as noted in the text) they do not depend on the local force of infection. In a sense, these are simply the latent parameters of the model, whereas policy makers need estimates of quantities that are closer to real-world observables. To convert the maps of protective duration to maps of (e.g.) mean number of infections averted to 30 days, and (more importantly) mean number of clinical incidence events averted to 30 days, requires that the end-user can run the Imperial mathematical model of malaria transmission calibrated to their local prevalence and then apply the intervention effect. Ideally, this could be done by the authors and presented in the manuscript, with some age-based stratification.

(At the risk of citing a politicised analogy, I would compare this to estimates of the transmission blocking potential of anti-covid interventions such as installation of ventilation in school classrooms and workplaces. In that case, simply quoting an estimate that e.g. a properly installed ventilation system will have 80% efficacy against covid transmission in the classroom, it might be more useful to compute how many cases per year in those children will be averted by the intervention. If the children are going to be infected anyway through life outside the classroom, then the impact of ventilation in the absence of widespread community control measures might be seen as rather underwhelming. See, eg. Ferguson et al. 2020's

conclusion that school closures alone only flatten the peak but not the final epidemic size. In the malaria context we really need to know what the intervention is buying us in terms of averted clinical cases to value the intervention, as I see it.)

(Remarks on code availability)

The code is as I would expect: entirely readable for someone familiar with modern statistical computing.

Reviewer #2

(Remarks to the Author)

The authors undertook an ambitious and important research question that will be difficult to address by assessing results of just a handful of studies. SP represents a remarkably resilient antimalarial, which has retained its role as a prophylactic in pregnancy, component of SMC, and now a mainstay of PMC (formerly IPTi). The study design is challenging, and the authors do an excellent job of pointing out the limitations. However, significant concerns about the validity of the modeled results exist because of all the limitations listed (differences in lab methods, effect of age, immunity, transmission intensity/seasonality, drug levels).

1) Introduction - please provide some additional clarity as to the results for some of the studies in the introduction.

- For the study in Uganda, line 92, the SP efficacy is reported versus the control arm. Is this in the absence of genotyping data?

- For the study in Tanzania, line 88, are these results genotype specific as well?

2) A major concern is the lack of incorporating transmission intensity as well as seasonality into the analysis, as this will significantly impact protective efficacy. Could the authors provide more detail as to the impact of not including this? Could a sensitivity analysis be done incorporating this as a factor?

3) What about the impact of ITNs as a covariate in the analyzed studies. Were all children under ITN coverage, insecticide resistance, etc...?

4) The authors note that the method of follow-up was microscopy, and this could vastly underestimate recurrence risk. They further state that the risk of recurrent parasitemia parallels clinical failure (line 273). However, more evidence to support this claim is needed. This is especially true in high intensity areas where children could acquire immunity very quickly. Please elaborate.

5) Please explain lines 155-157 - this needs more explanation/justification

6) Line 163 - was the term substantially used purposefully (versus significantly)? If there are statistically significant results in any of the comparisons, it would be good to note that.

7) It seems to be too much of a stretch to model the impact of GEG genotype if it was based on one partial study, shorter follow-up. Would eliminate that assessment due to a lot of uncertainty in the conclusions

8) Line 179-181 - this is an interpretation, and should be moved to the discussion

9) I would also strongly consider downplaying the drug analysis, as it appears to be based on only one study. In addition, please comment on the potential role of underdosing in young children, as this has been shown by Barnes, et al. and could confound results in the group getting PMC.

10) The interpretation of why the duration of protection of SPAS is longer than SP in lines 289-295 provides a great deal of conjecture, and not clear why misclassification would be more likely in the SP arm. Please explain in more detail.

11) Please provide more justification/detail on the validation studies, as no genotyping data from those exact studies was available. How likely is it that the data used from other sources was reflective of the status in those trials?

12) Readers may want more explanation as to why studies on SPAQ as SMC were not included (unless I misunderstood this). please explain

(Remarks on code availability)

Reviewer #3

(Remarks to the Author)

The manuscript is appropriate for acceptance with minor edits. This manuscript will contribute to the field of surveillance of antimalarial drug efficacy, its importance and approach to describe it. This study shows that there is a shorter duration of Sulfadoxin-Pyrimethamine (SP) protection against parasite carrying more mutations in Africa and that the duration of protection increases with SP combination therapies/treatments, based on the genotype data of Pfdhps gene from more than 16K samples in W. and E. Africa. The figures and tables are well defined but can be improved by providing high resolution

figures. The study uses multiple available tools, publicly available databases, and custom scripts. The work meets expected standards in the field of anti-malarial drug protective efficacy genomic surveillance.

Minor points/questions/suggestions:

1. General,

a. Information on demographics of 1639 samples can help to understand the differences between results from different locations.

b. In Table 1 it is not clear if the percentage is shown only for the dominant allele or for the only alternate allele (rest being reference/susceptible – AKA). What is the percentage of other alleles?

(Remarks on code availability)

Version 1:

Reviewer comments:

Reviewer #1

(Remarks to the Author)

I am happy to confirm that the authors have thoroughly addressed the comments from my initial review. I have no additional comments so am please to recommend the manuscript for publication.

(Remarks on code availability)

Reviewer #2

(Remarks to the Author)

I thank the reviewers for their thorough and transparent response to all comments provided. I believe the additions will enable the reader to understand the studies limitations better, and also have confidence in the soundness of its results, as presented. I have no further issues/concerns.

(Remarks on code availability)

We sincerely thank the reviewers and the editor for the thorough feedback and constructive suggestions. We have made substantive changes to the manuscript, including all recommended sensitivity analyses and additional outcomes, which we believe have significantly improved its quality and clarity. We hope that these revisions have addressed all concerns and that the manuscript is now suitable for publication.

Reviewer #1 comments	Response
In the manuscript entitled "Impact of dhps mutations on sulfadoxine-pyrimethamine protective efficacy and implications for malaria chemoprevention" the authors present the results of a statistical analysis to estimate the relationship between dhps mutation type and duration of protective efficacy against malaria re-infection. Overall the study is of high quality, by which I mean that the objective is valuable and due diligence and effort has been applied to the data preparation and model development; also, the statistical methods are consistent with best practice for hierarchical modelling in the contemporary literature. I have two comments/questions that I would urge the authors to address in their revisions. The first is critical to establishing the validity of the model-based approach, while the second is concerned with facilitating clear interpretation of results for policy makers.	We thank the reviewer for the valuable comments and suggestions. We have revised the manuscript to include sensitivity analyses to strengthen the validity of our approach, as well as providing additional outputs that have more direct policy implications. Please see details below on how we have addressed these, on a point-by-point basis.
The dataset used in this analysis come from a type of drug efficacy trial in which there is no untreated arm from which to estimate the background force of infection. For this reason the force of infection is treated as a latent model parameter that takes a constant value over the study duration at each site; this constant force of infection at each site is assigned a semi-informative prior in a Bayesian framework based on a mathematical model for the relationship between prevalence and the entomological inoculation rate. As acknowledged in the Discussion, the constant rate assumption is quite limiting, as	As the reviewer points out one major limitation is that in the main analysis there were no control arms due to the nature of the studies. The pooling of these studies is what enables disaggregation of drug effects and site-specific incidence, which was assumed to be constant over time. We have now justified this in the methods (first paragraph under "Analysis methods"): "Incidence of infection was assumed to be constant over the 42 days follow-up, due to the challenge of identifying fluctuating transmission effects in the absence of a control group." In the validation analysis we used two IPTi trials which included control groups, and this enabled us to account for fluctuating transmission during the study follow-up. We have followed the reviewer's excellent suggestion and repeated the validation analysis, this time withholding the control arms. This is now an additional paragraph in the methods section under "Validation":

any variations of transmission intensity during the study week will confound the shape estimates of the efficacy functions. I would guess that a further limitation is the assumption of no heterogeneity in the exposure risk of individuals; certainly the mathematic model used to estimate the priors for the force of infection at each site considers both heterogeneity between individuals and an age-dependence of the biting age. On the other hand, the comparison models fitted to the two validation datasets for which there are control arms are constructed with time-varying incidence rates, which makes the agreement between the output curves and the data difficult to assess. Further sensitivity analyses would therefore be appreciated. For instance, examining how the latent incidence rate estimates for the validation datasets compare with the observations when their control arms are withheld (under a time-constant force of infection model).

“As an additional sensitivity analysis, we repeated the validation analysis with a time-constant force of infection and compared the estimated incidence rate in the IPTi trial by including and then excluding the placebo-controlled arm. We further compared the estimated duration of protection with that obtained from the analysis using time-varying incidence, and that expected using the protection parameters from the main analysis of the TES trials.”

We found that the results on the duration of protection estimated from the IPTi trial single chemoprevention arm was similar to that expected given the parameters from the main analysis (as well as frequency of genotypes and incidence expected for the IPTi trial settings). We have added a Supplementary Table 5 with the results of all validation analyses, as well as the following in the results section (3rd paragraph under “Duration of protection against different genotypes”):

“Similar estimates for protection were obtained when a time-constant force of infection was assumed across follow-up (Supplementary Table 5). Withholding the control group from the analysis resulted in slightly shorter protection for the trial in Mozambique (20.7 days) and slightly longer protection for the trial in Tanzania (14.9 days).”

Note that these estimates are very close to those expected using the parameters from the main analysis.

We thank the reviewer for the suggestion on testing the assumption of heterogeneity. We have added a new sensitivity analysis that includes heterogeneity in risk of exposure when pooling the TES studies (results shown in Supplementary Table 7). Accounting for heterogeneity in risk made little difference to the estimates of protection.

In the methods section (3rd paragraph under “Analysis Methods” we added:

“As a sensitivity analysis, we explored the effect of including heterogeneity in risk of transmission in the main analysis. To account for variation in risk between individuals, three risk groups were used, partitioned using Gaussian quadrature (Supplementary Table 7).”

In the Results section (last paragraph under “Duration of protection against different genotypes”) we added:

“Accounting for heterogeneity in the risk of transmission resulted in similar estimates of mean duration of protection across all drugs (Supplementary Table 7).”

Regarding the presentation of final results, while the duration of protection metrics have a clear mathematical explanation I fear that they may not be meaningful to potential end-users in the policy sphere, since (as noted in

We thank the reviewer for this suggestion that can improve the utility of these results to inform policy. As suggested by the reviewer, we have converted the Africa-wide maps of protective efficacy and duration of protection to maps of clinical cases averted following a single-dose of SP. This is shown in Figure 7 in the main text and Supplementary Figure 5. We also present incidence with and without a dose

the text) they do not depend on the local force of infection. In a sense, these are simply the latent parameters of the model, whereas policy makers need estimates of quantities that are closer to real-world observables. To convert the maps of protective duration to maps of (e.g.) mean number of infections averted to 30 days, and (more importantly) mean number of clinical incidence events averted to 30 days, requires that the end-user can run the Imperial mathematical model of malaria transmission calibrated to their local prevalence and then apply the intervention effect. Ideally, this could be done by the authors and presented in the manuscript, with some age-based stratification.

(At the risk of citing a politicised analogy, I would compare this to estimates of the transmission blocking potential of anti-covid interventions such as installation of ventilation in school classrooms and workplaces. In that case, simply quoting an estimate that e.g. a properly installed ventilation system will have 80% efficacy against covid transmission in the classroom, it might be more useful to compute how many cases per year in those children will be averted by the intervention. If the children are going to be infected anyway through life outside the classroom, then the impact of ventilation in the absence of widespread community control measures might be seen as rather underwhelming. See, eg. Ferguson *et al.* 2020's conclusion that school closures alone only flatten the peak but not the final epidemic size. In the malaria context we really need to know what the intervention is buying us in terms of averted clinical cases to value the intervention, as I see it.)

SP and cases averted separately for ages 0 to 2 and 0 to 5 (see Supplementary Figure 5).

We have added the following paragraph in the methods section (last paragraph in "Analysis Methods":

"We used prevalence estimates in children aged 2 to 10 for the year 2020 obtained from the Malaria Atlas Project at 5km x 5km resolution.⁵⁵ The relationship between parasite prevalence in 2 to 10 year olds and incidence in 0 to 2 year olds, obtained from an agent-based malaria transmission model,^{53,54} was then used to infer incidence in each 5km x 5km pixel across Africa. The 30-day protective efficacy was then applied to these incidence metrics to estimate the number of clinical cases aged 0 to 2 per population after a single SP dose. The same process was repeated with incidence in 0 to 5 year olds to compare impact in these two age groups. To calculate mean clinical cases averted per dose for each country in areas of moderate to high transmission, we weighted the values by the population in each 5km x 5km pixel, using 2020 WorldPop population data.⁵⁸"

In the last paragraph of the results section:

"Using our estimated protective efficacy profiles against each genotype, we used previously published estimates of *dhps* genotype frequencies³⁵ to predict the SP 30-day protective efficacy against any new infection and the median duration of protection (Figure 6 and Supplementary Figure 4). Across sub-Saharan Africa, 30-day protective efficacy varied from 59.3% to 91.5%, and the median duration of protection ranged from 17.2 to 37.2 days. We estimated that this protective efficacy corresponded to a mean of 4.7 clinical cases averted per 100 children aged 0 to 2 following a single-dose in areas of moderate-to-high transmission ($PRPf_{2-10} \geq 10\%$) (Figure 7, Supplementary Figure 5). Similar reductions were estimated in the 0 to 5 year age group (Supplementary Figure 5). The highest mean number of clinical cases averted per 100 per dose was estimated for Liberia (8.9), followed by Benin (8.5), Sierra Leone (7.3) and Democratic Republic of the Congo (6.9)".

Please note that within the wider grant activities focussed on perennial malaria chemoprevention (PMC) within our team at LSHTM and with international partners (Plus Project), a decision-support tool is being developed to help policy makers make decisions around PMC with SP. This deals with exploring the effectiveness and cost-effectiveness of different PMC delivery schedules and coverage across sub-Saharan Africa. We do not want to pre-empt these results and have, therefore, limited our outcomes to cases averted per population after a *single* SP dose.

1 (Remarks on code availability): The code is as I would expect: entirely readable for someone familiar with modern statistical computing.	We thank the reviewer for checking the code, and for the feedback.
--	---

Reviewer #2 comments	Response
The authors undertook an ambitious and important research question that will be difficult to address by assessing results of just a handful of studies. SP represents a remarkably resilient antimalarial, which has retained its role as a prophylactic in pregnancy, component of SMC, and now a mainstay of PMC (formerly IPTi). The study design is challenging, and the authors do an excellent job of pointing out the limitations. However, significant concerns about the validity of the modeled results exist because of all the limitations listed (differences in lab methods, effect of age, immunity, transmission intensity/seasonality, drug levels).	We thank the reviewer for the comments and suggestions as well as acknowledging the challenges in the lack of suitable data and the need for a study design that allows inference of SP effects from available sources. We have explored some of the stated confounding factors in the analyses. In all analyses, including the main analyses we have accounted for transmission intensity in the study site by fitting a separate parameter for the force of infection. In the validation analyses using IPTi trial data, we have also fitted separate force of infection parameters for each week to account for seasonality/fluctuations in transmission during follow-up (See Figure 4 and Supplementary Table 5). Investigating seasonality was only possible for data which included a control group, otherwise protection and time-varying incidence parameters are not identifiable. The findings from the IPTi trials used for validation closely matched what we expected from the main analysis, increasing the confidence in our results. We have also explored the impact of drug levels, where this information was available, and did not observe any confounding effects (See Supplementary Text 2). We have added the following sentence in the third paragraph of the discussion, relevant to the impact of immunity on protective efficacy: “However, more evidence is needed to assess whether protective efficacy against any infection is proportional to efficacy against clinical infection, particularly in high-transmission areas with high rates of acquired immunity and highly prevalent asymptomatic parasitaemia.”
1) Introduction - please provide some additional clarity as to the results for some of the studies in the introduction.  - For the study in Uganda, line 92, the SP efficacy is reported versus the control arm. Is this in the absence of genotyping data? - For the study in Tanzania, line 88, are these results genotype specific as well? 	We thank the reviewer for the suggestion to improve clarity of the introduction paragraph detailing current evidence from IPTi trials. The two trials mentioned in this section do not report (and possibly did not measure) drug resistance markers. Indeed, the statements about the prevalence of these markers are informed by separate studies done in the same districts and years as the IPTi trials. To clarify this, we have made the following edits in the paragraph: “Trials of IPTi and IPT in pregnant women (IPTp) suggest reduced effectiveness in areas with high prevalence of SP resistance markers, though data on resistance markers

	typically come from separate studies. In Korogwe, Tanzania (2004-2008), there was no significant protective efficacy conferred by IPTi-SP after 21 days.^{12,14,15} Evidence from a trial done in the same area and period as the IPTi trial, showed that the GEG genotype was present in approximately half of the samples collected on the day of enrolment¹³. Similarly, IPTp-SP studies reported an association between the dhps GEG genotype and loss of protection from infection,^{16,17} and low birthweight.¹⁸ An IPTi trial conducted in Uganda, reported an SP efficacy of just 7% against clinical malaria compared to the control arm.¹⁹ In the same district, a trial found almost all episodes of malaria on the day of recruitment were with parasites carrying the GEA genotype with no 581G.²⁰"
2) A major concern is the lack of incorporating transmission intensity as well as seasonality into th analysis, as this will significantly impact protective efficacy. Could the authors provide more detail as to the impact of not including this? Could a sensitivity analysis be done incorporating this as a factor?	We thank the reviewer for this comment and suggestions. To clarify, in the main analysis pooling together TES data we have incorporated the effect of transmission intensity as an explicit parameter which is allowed to differ in each trial site. We did not include the effect of transmission fluctuating over time during a single trial, or seasonality in the site, however, most trials had relatively short duration (up to 6 weeks of follow-up). None of the TES trials included a control group as they were done at a time when SP was used as a first line treatment. The pooling of these studies together is what enables disaggregation of drug effects and site-specific incidence, which was assumed to be constant over time. The impact of this is acknowledged as a limitation in the 2nd to last paragraph of the discussion: “One of the limitations of this study is the lack of control groups in the trial settings. This means the true underlying transmission rates are not directly measured and assumed to be constant. In the absence of a control group, a long follow-up is needed to isolate drug effects, and is particularly challenging in settings with fluctuating or seasonal transmission, which can result in inaccurate estimates of protection.” We have now justified this in the methods (first paragraph of analysis methods): “Incidence of infection was assumed to be constant over the 42 days follow-up, due to the challenge of identifying fluctuating transmission effects in the absence of a control group.” However, in the validation analysis we used two IPTi trials which included control groups, and this enabled us to account for fluctuating transmission. In the revised version, we have repeated the validation analysis, this time withholding the control arms, as suggested by another reviewer. This is now an additional paragraph in the methods section under “Validation” (2nd paragraph):

	“As an additional sensitivity analysis, we repeated the validation analysis with a time-constant force of infection and compared the estimated incidence rate in the IPTi trial by including and excluding the placebo-controlled arm. We further compared the estimated duration of protection with that obtained from the analysis using time-varying incidence, and that expected using the protection parameters from the main analysis of the TES trials.” We found that our estimated duration of protection from the analysis of TES trials was able to produce an excellent prediction of the results of an independent IPTi trial (allowing for local frequency of genotypes and incidence). We have added a Supplementary Table 5 with the results of all validation analyses, and edited the results section (4th paragraph under “Duration of protection against different genotypes”). See also our response to reviewer 1 above about the additional sensitivity analysis removing control arm data from the validation datasets.
3) What about the impact of ITNs as a covariate in the analyzed studies. Were all children under ITN coverage, insecticide resistance, etc...?	We thank the reviewer for this suggestion. The coverage of ITNs may result in lower or higher estimated baseline incidence but we do not expect it to affect the findings on the duration of protection by drug and against each strain, given the randomized design of the studies. The only scenario in which ITN coverage may confound findings on protective efficacy is if coverage varied during follow-up, or if it is different by drug group where more than one drug group was examined. Data on intervention coverage are not available in the TES studies included in the main analysis, and we have no reason to believe that ITN coverage may fluctuate systematically over follow-up, or between drug arms. In fact, the IPTi trial in Tanzania by Gosling et al. used here for validation has also recorded a remarkably similar bednet coverage in the placebo and in the SP arms (87.8% vs 86.2%, respectively). Similarly, in the study by Nahum et al. used in the main analysis showed that bednet coverage was also very similar between treatment groups (ranged between 72.1% and 81.1%).
4) The authors note that the method of follow-up was microscopy, and this could vastly underestimate recurrence risk. They further state that the risk of recurrent parasitemia parallels clinical failure (line 273). However, more evidence to support this claim is needed. This is especially true in high intensity areas where children could acquire immunity very quickly. Please elaborate.	We thank the reviewer for the comment and insights. Samples were actively tested by microscopy and then confirmed by PCR. We agree with the reviewers, that PCR may have detected a few more breakthrough infections that were suppressed by the immune system and were not detected by microscopy. Using a more sensitive method of diagnosis (e.g. PCR) would also increase the estimated baseline incidence, but may not necessarily change conclusions on the duration of drug protection. Microscopy testing is the method recommended by WHO for use in TES studies and for quantification of treatment failure during follow-up. Therefore, microscopy positive infections (confirmed by PCR) are the only available data from such studies. This outcome may be more relevant to clinical outcomes, as low density parasitaemia undetected by

	microscopy is unlikely to result in a symptomatic malaria episode. In our analysis we make the assumption that within any setting, a given percentage reduction in incidence of infection would result in a similar percentage reduction in clinical disease, an assumption supported by a systematic review of IPTi trials (Esu et al. 2021). However, as suggested by the reviewer, we have added the following in the 3rd paragraph of the Discussion section to highlight the lack of robust evidence related to the comparison of clinical and parasitological outcomes: “However, more evidence is needed to assess whether protective efficacy against any infection is proportional to efficacy against clinical infection, particularly in high-transmission areas with high rates of acquired immunity and highly prevalent asymptomatic parasitaemia.”
5) Please explain lines 155-157 - this needs more explanation/justification	We thank the reviewer for the comment and points for clarification. We have edited the text to improve clarity: “The mean duration of protection against new infection was calculated using the estimated shape and scale parameters of the Weibull survival curve (Supplementary Text 1). In our modelling approach we model partial protection, rather than using a step function where someone is either protected or not on a particular day. Partial protection is expressed as a probability of protection over time since dose and this curve is constant across settings, irrespective of transmission. This concept has been explained in more detail in a previous publication (see Figure 1 in https://doi.org/10.1371/journal.pmed.1004376).
6) Line 163 - was the term substantially used purposefully (versus significantly)? If there are statistically significant results in any of the comparisons, it would be good to note that.	Thank you for the suggestion on improving clarity of this statement. As we used a Bayesian inference method we had initially omitted p-values, but have now included them as an additional column in Table 2 with some explanation on how p-values were computed: “p-values are calculated as the proportion of the posterior sample differences that are less than or greater than 0 (two-tailed test).” We have also edited the text in the 2nd paragraph under “Duration of protection against different genotypes” in the Results section to report the estimated p-values. The sensitive genotype (dhps AKA) was used as the reference category.

7) It seems to be too much of a stretch to model the impact of GEG genotype if it was based on one partial study, shorter follow-up. Would eliminate that assessment due to a lot of uncertainty in the conclusions	We thank the reviewer for the comment, and understand their concern that only one study (in the main analysis) covers the GEG genotype. We have limited information on the effect of the GEG genotype as SP was quickly withdrawn as a first-line treatment in areas with the 581G mutation. However, we feel it is appropriate to include all available data for completeness, including this study. Uncertainty in the GEG estimate is reflected in the relatively wide credible intervals in Table 2: 8-22 days). In addition, for areas with low levels of resistance, a 28-day follow-up study may not provide enough information to disaggregate the drug protection effect. However, the 28-day follow-up may be sufficient in the absence of a control group if the parasites have high levels of resistance, which we expect to be the case for the GEG genotype. The validation of our estimate of the GEG protective efficacy against the Tanzanian IPTi trial data, a separate dataset in a site where GEG was prevalent, is reassuring since we were able to predict the results of this independent study well. This is the extract from the discussion that addresses the lack of GEG data (second to last paragraph of discussion): “More data are needed to inform our parameter estimates, particularly on the more resistant dhps GEG genotype. This genotype was only covered by one of the included studies, which had a shorter follow-up of 28 days”.
8) Line 179-181 - this is an interpretation, and should be moved to the discussion	Thank you for pointing this out. We have removed this interpretation statement from the results, as suggested. It is discussed in the 5th paragraph of the discussion.
9) I would also strongly consider downplaying the drug analysis, as it appears to be based on only one study. In addition, please comment on the potential role of underdosing in young children, as this has been shown by Barnes, et al. and could confound results in the group getting PMC.	We thank the reviewer for this suggestion. In the methods section we mention that sulfadoxine and pyrimethamine drug concentration data were available in only one study: “Drug concentration data on sulfadoxine and pyrimethamine on day 7 following drug administration were only partially available for one study.^{25,34}” We have now made sure that this is also mentioned in the results section: “Neither day 0 drug concentrations nor initial parasite density were associated with time to reinfection (Supplementary Text 2), though these were only available from a single trial.^{25,34}” Other than this statement in the results section, the remaining analysis is in the supplement which we feel does not overplay the drug analysis. As suggested, we have also included in the discussion a comment about the effect of appropriate dosing, and relevant considerations (second to last paragraph of Discussion): “Underdosing in young children is associated with treatment failure⁴⁴ and was not explored in this analysis. We expect that this effect is the same across studies, except in those using different age groups, where the

	genotype effects on protection may be harder to distinguish from age effects. However, our accurate prediction of the IPTi trial results despite differences in age groups, suggests that this confounding effect is minimal.”
10) The interpretation of why the duration of protection of SPAS is longer than SP in lines 289-295 provides a great deal of conjecture, and not clear why misclassification would be more likely in the SP arm. Please explain in more detail.	We thank the reviewer for the comment and suggestion. The model we employ only seeks to estimate the duration of protection from new infections with parasites acquired after the drug dose. All known episodes of PCR-confirmed recrudescence after day 0 are censored. In reality however, parasite genotypes may be present at the time a drug is given but not be detected by microscopy and PCR, due to low-density parasitaemia or minority infections. During the PCR amplification process, if a particular strain has <10%-20% prevalence in a sample, it may not be detected. If this infection is then detected after day 0, it may be misclassified and analysed as a “new infection”, rather than be censored (as a recrudescence infection). The AS component of SPAS provides no long-term protection but is effective at clearing existing parasites, hence more likely to clear those minority infections on day 0. This may explain why the estimated duration of protection provided by SPAS is longer than SP alone. To some extent we have accounted for this potential misclassification in the SP group by censoring all parasite recurrence in the first week as “recrudescence” (for all arms), though it is still likely that protection in the SP-only group is partly underestimated due to this misclassification problem, expected to inflate the number of apparent “new infections”. Hence we consider the SPAS protection estimates to be more reliable and less biased. To address the reviewer’s concern, we have edited the (5th) paragraph in the Discussion section to capture and describe these arguments in a clearer manner, supported by the addition of relevant citations.
11) Please provide more justification/detail on the validation studies, as no genotyping data from those exact studies was available. How likely is it that the data used from other sources was reflective of the status in those trials?	Unfortunately genotype data are not available from the IPTi study in Tanzania. This IPTi trial data was from Korogwe (Tanga region) between Dec 7, 2004, and May 1, 2008. The genotyping data was from a TES study by Gesase et al. 2009 which was conducted between July-August 2006 (so half-way through the IPTi trial) in Hale Health Centre, Tanga Region, situated 30 km south of Korogwe. Given the proximity of the two sites, we expect that the genotype frequencies will be similar, supported by other work suggesting that resistance measures are similar within a 300km spatial scale (https://doi.org/10.1038/s41598-017-06708-9). In the case of the IPTi trial in Mozambique (Macete et al.), the genotype frequency information comes from samples from symptomatic cases in the placebo group of the IPTi trial. The frequencies from the placebo group should represent those in the general parasite population, in the absence of any drug selection pressure. We realise that

	we had not mentioned that these samples come from the same trial, so have now corrected this and added any missing information. See edits below (1st paragraph under “Validation” section of the Methods): “Placebo-controlled IPTi-SP trial data^{12,21} were used for validation of our results, and were obtained from trial investigators or digitized from published Kaplan-Meier survival curves for both treatment and control arms after the first dose of SP in the trial and before any further doses. Where no dhps genotype data were available from the IPTi trial participants, we used data on the frequency of SP-resistance markers in that area and year from other sources. In the case of the IPTi trial in Mozambique,²¹ the genotype frequency information was obtained from samples collected from symptomatic malaria cases in the placebo group of the IPTi trial,²² representing population frequencies in the absence of drug selection. In the case of the IPTi trial in Tanzania, frequency estimates were obtained from samples collected in a health facility within 30km of the IPTi implementing district, and during the same time as the IPTi trial (in 2006). Using both control and treatment arms, we estimated weekly clinical incidence and a Weibull survival curve for protective efficacy using RStan as above. We then predicted new infections given the estimated incidence and local frequencies of resistance genotypes^{13,22} and compared the predicted duration of SP protection with that estimated in the observed IPTi data used for validation.” Please also note that we have added sensitivity analyses around the validation datasets which are in the new Supplementary Table 5.
12) Readers may want more explanation as to why studies on SPAQ as SMC were not included (unless I misunderstood this). please explain	We thank the reviewer for the suggestion. The protection parameters estimated from this study are directly relevant to SP chemoprevention for PMC rather than SMC. However, the approach and modelling framework we are using can be applied to other chemoprevention drugs. This is mentioned in the last paragraph of the discussion: “This approach establishes a valuable methodology which can be applied across all epidemiological settings where resistance profiles have been characterised, and can be applied to other treatment regimens including SPAQ and DP, by quantifying the effects of relevant markers such as mdr 1 and crt.” Although we included data from an SPAQ arm in the study done by Bell et al. in Malawi, and found a very long duration of SPAQ protection, we chose not to make any strong statements about SMC efficacy, as this analysis was done opportunistically. A thorough review of the literature for SPAQ studies would be needed to assess this properly, which would require a large amount of additional work. To that end, we are collaborating on a separate study looking to answer this exact question by conducting

	a review of a large number of publications and combining all evidence of SPAQ protective efficacy in the presence of different dhps and mdr mutations.
--	--

Reviewer #3 comments	Response
The manuscript is appropriate for acceptance with minor edits. This manuscript will contribute to the field of surveillance of antimalarial drug efficacy, its importance and approach to describe it. This study shows that there is a shorter duration of Sulfadoxin-Pyrimethamine (SP) protection against parasite carrying more mutations in Africa and that the duration of protection increases with SP combination therapies/treatments, based on the genotype data of Pfdhps gene from more than 16K samples in W. and E. Africa. The figures and tables are well defined but can be improved by providing high resolution figures. The study uses multiple available tools, publicly available databases, and custom scripts. The work meets expected standards in the field of anti-malarial drug protective efficacy genomic surveillance.	We thank the reviewer for the valuable feedback. We are providing higher resolution figures, in line with journal guidelines.
Minor points/questions/suggestions: 1. General, a. Information on demographics of 1639 samples can help to understand the differences between results from different locations.	Thank you for the suggestion on improving the demographic information provided for the different trials. We acknowledge that demographic differences between sites, and other heterogeneity between trials, may confound our results. The only demographic variable that was available for all studies was age. We had previously included the age range of each study in Table 1. We have now added the mean age in the table for all studies. It is worth highlighting that the genotype-specific drug profiles are remarkably similar between sites despite any demographic differences.
b. In Table 1 it is not clear if the percentage is shown only for the dominant allele or for the only alternate allele (rest being reference/susceptible – AKA). What is the percentage of other alleles?	We thank the reviewer for this suggestion to improve clarity of this table. We have now ensured that the column reporting percentages for the frequency of each allele adds up to 100%. There were only two exceptions. The first was the Nahum et al. study, for which we could not retrieve the individual level data, though the tables (2 and 3) in the original 2009 publication indicate that the majority of genotypes circulating are dhps GKA. The other exception was the Bredenkamp et al. , 2001 study. For this study although the parasitological outcomes were available at the individual-level, the dhps genotyping results were not.